

# Low oxygen alters mitochondrial function and response to oxidative stress in human neural progenitor cells

Yury M. Lages[1], Juliana M. Nascimento[2], Gabriela A. Lemos[3], Antonio Galina[3], Leda R. Castilho[4] and Stevens K. Rehen[1,2]

[1] Institute of Biomedical Sciences, Federal University of Rio de Janeiro, Rio de Janeiro, RJ, Brazil
[2] IDOR, D'Or Institute for Research and Education, Rio de Janeiro, RJ, Brazil
[3] Institute of Medical Biochemistry Leopoldo De Meis, Federal University of Rio de Janeiro, Rio de Janeiro, RJ, Brazil
[4] COPPE, Chemical Engineering Program, Federal University of Rio de Janeiro, Rio de Janeiro, RJ, Brazil

Corresponding author
Stevens K. Rehen,
srehen@lance-ufrj.org

## ABSTRACT

Oxygen concentration should be carefully regulated in all living tissues, beginning at the early embryonic stages. Unbalances in oxygen regulation can lead to cell death and disease. However, to date, few studies have investigated the consequences of variations in oxygen levels for fetal-like cells. Therefore, in the present work, human neural progenitor cells (NPCs) derived from pluripotent stem cells grown in 3% oxygen (v/v) were compared with NPCs cultured in 21% (v/v) oxygen. Low oxygen concentrations altered the mitochondrial content and oxidative functions of the cells, which led to improved ATP production, while reducing generation of reactive oxygen species (ROS). NPCs cultured in both conditions showed no differences in proliferation and glucose metabolism. Furthermore, antioxidant enzymatic activity was not altered in NPCs cultured in 3% oxygen under normal conditions, however, when exposed to external agents known to induce oxidative stress, greater susceptibility to DNA damage was observed. Our findings indicate that the management of oxygen levels should be considered for *in vitro* models of neuronal development and drug screening.

## INTRODUCTION

By describing events that alter mitochondrial metabolism and facilitate tumor formation, *Warburg (1956)* shed light on the importance of oxygen concentrations for cellular health. Accordingly, matching the *in vitro* conditions of a given cell type to the respective *in vivo* oxygen concentration has become a relevant issue that accompanies the growing number of applications of human pluripotent stem cells, which are particularly relevant for modeling fetal and/or neurological disorders.

Mitochondrial function and oxygen metabolism not only determine aspects of neural development (*Li et al., 2004*), but they are also strongly implicated in the etiology and

progression of brain disorders, including Parkinson's disease, Alzheimer's disease, and schizophrenia (*Paulsen et al., 2013*; *Yan, Wang & Zhu, 2013*). Impairment of mitochondrial function or the redox state may be especially problematic for highly metabolically demanding neurons. Mismanagement of these processes is massively problematic, negatively impacting energy metabolism, neurochemical signaling and/or synaptic plasticity, and emergent cognitive processes of these functions (*Cheng, Hou & Mattson, 2010*; *Janc & Muller, 2014*; *Tait & Green, 2012*).

Despite the well-recognized relationship between oxidative metabolism and the onset of neural disorders (*Paulsen et al., 2013*; *Yan, Wang & Zhu, 2013*; *Carreau et al., 2011*), few studies have focused on analyzing changes occurring at atmospheric oxygen concentrations (i.e., 21% $O_2$ (v/v); typical levels in cell culture, normoxia), compared with physiological levels (3% $O_2$ (v/v)). Studies carried out on murine neural progenitor cells (NPCs) have considerable differences in proliferation, death, and differentiation (*Bae et al., 2012*; *Chen et al., 2007*; *Rosafio & Pellerin, 2014*; *Ross et al., 2012*; *Stacpoole et al., 2011*; *Studer et al., 2000*). These studies have shown unexpected deviations in cell fate, including altered relative proportions of neuronal and glial populations (*Chen et al., 2007*; *Stacpoole et al., 2011*; *Studer et al., 2000*).

Studies that specifically address the impacts of oxygen levels on the metabolic behavior of NPCs are still rare. Recent reports have described increased dispersion of mitochondria as well as modifications in mitochondrial efficiency and reactive oxygen species (ROS) production of rat neurons grown under 1–5% $O_2$ (*Tiede et al., 2011*). In addition, Tiede and colleagues (*2011*) have reported increased cell death in physiological oxygen concentrations (physioxia (*Rosafio & Pellerin, 2014*)) when NPCs are exposed to viral infection proteins; however, their study did not elucidate the cause of the alterations. Therefore, the aim of this study was to compare NPCs grown in physioxia and normoxia (3% and 21% (v/v) $O_2$, respectively) in terms of growth kinetics, glycolytic metabolism, mitochondrial content, mitochondrial membrane potential ($\Delta\Psi_M$), oxygen uptake, ATP production, ROS production, and antioxidant enzymatic activity.

## METHODS

### Differentiation of human embryonic stem cells into NPCs

To generate NPCs, human embryonic stem cells BR-1 (*Fraga et al., 2011*) (kindly provided by Prof. Lygia Pereira, São Paulo University - USP) were grown on polystyrene plates (TPP, Switzerland) covered with Matrigel (BD Biosciences, Franklin Lakes, NJ, USA) in StemPro medium containing 8 ng/mL basic fibroblast growth factor and 0.1 mM $\beta$-mercaptoethanol (all from Thermo Fischer Scientific, Waltham, MA, USA). After propagation, the cells were differentiated as neural cells using inhibitors of bone morphogenetic protein (Noggin; R&D Systems, USA), and transforming growth factor-beta (SB431542; Tocris Bioscience, Bristol, UK) (*Chambers et al., 2009*). At this moment, cells showed morphology and expression of markers consistent to those of neural progenitor cells Figs. S2A and S2B.

## Cultivation of NPCs in a physiological environment

NPCs were grown in an environment containing 3% oxygen (physioxia) in an oxygen control chamber (ProOx model C21; BioSpherix, Parish, NY, USA). This equipment was kept at 37 °C and 5% $CO_2$, and 3% $pO_2$ was established by a $N_2$-controlled injection and monitored by an external probe (Mettler Toledo, Colombus, OH, USA).

## Growth kinetics

Cellular growth under different oxygen conditions was evaluated for 18 days (3 passages of 6 days). A total of $6 \times 10^5$ NPCs/mL were plated into 24-well tissue culture plates. Each day, cells from two wells of each condition were detached with Accutase (Millipore, Darmstadt, Germany) and counted in a Neubauer chamber.

## Glucose and lactate measurements

Glucose and lactate concentrations were determined using a YSI-2700 biochemistry analyzer (Yellow Springs Instruments, Yellow Springs, OH, USA). This measurement is based on quantification of hydrogen peroxide generated upon reaction of these organic molecules catalyzed by glucose or lactate oxidases immobilized on membranes.

## Immunostaining assays

NPCs were seeded in 96-multiwell μClear dishes (Greiner, Austria) covered with 2.5 μg/mL laminin (Sigma-Aldrich, USA). After 6 days, these cells were fixed with 4% paraformaldehyde (Sigma-Aldrich, USA) in phosphate-buffered saline for 30 min.

Then, the cells were treated with 0.5% Triton X-100 (Sigma-Aldrich, St. Louis, MO, USA), blocked with 5% bovine serum albumin (Sigma-Aldrich, St. Louis, MO, USA), incubated with the following primary antibodies: rabbit anti-human-histone H2A (H2A.X) (1:100; Cell Signaling, USA), mouse anti-Nestin (1:100; Chemicon, Temecula, CA, USA), mouse anti-PSA-NCAM (1:100; Millipore, Germany) and rabbit anti-Tbr2 (1:100; Millipore, Darmstadt, Germany). Subsequently, samples were incubated with the following secondary antibodies: goat anti-rabbit AlexaFluor 488 IgG (1:400; Thermo Fischer Scientific, Waltham, MA, USA) and goat anti-mouse Alexa Fluor 594 IgG (1:400; Thermo Fischer Scientific, Waltham, MA, USA). Nuclei were stained with 0.5 μg/mL $4'$-6-diamino-2-phenylindole (DAPI).

Regions of interest were visualized and identified, and the immunofluorescence emission of the cells was quantified using an Operetta high content analysis system and Harmony software (PerkinElmer, Waltham, MA, USA). In these experiments, three technical replicates of each biological replicate (N) were performed.

Twenty-five fields per well were chosen randomly. An average of 3,125 nuclei per well were analyzed. The high-content and high-throughput screening platforms used herein allow the evaluation of different samples in a large scale, with simultaneous identification of several parameters using automated fluorescence microscopy, using specific markers for different proteins (*Li, 2014*).

## Mitochondrial content and $\Delta\Psi_M$ quantification assays

Measurement of the mitochondrial mass of NPCs was performed using 0.3 μM Mitotracker DeepRed FM (Thermo Fischer Scientific, Waltham, MA, USA), a dye that integrates into active mitochondria (568-nm excitation and 675-nm emission). The $\Delta\Psi_M$ was estimated by cationic staining with 1.6 μM JC-1 (Thermo Fischer Scientific, Waltham, MA, USA) (488-nm excitation). This dye exists as a monomer at low concentrations, with fluorescence emission at 525 nm (shown here in green). As it accumulates in the mitochondria, which is membrane potential-dependent, the dye forms aggregates that exhibit a maximum emission at 590 nm (shown here in yellow). The ratio of aggregate to monomer concentration can be used as a measurement of $\Delta\Psi_M$ (*Reers, Smith & Chen, 1991*).

MitoTracker and JC-1 dyes, diluted in Dulbecco's modified Eagle's medium/F12 (Thermo Fischer Scientific, Waltham, MA, USA), were applied to NPCs for 40 min at 37 °C. Fluorescence emission readings were performed in a controlled 5% $CO_2$ and 37 °C environment. Hoechst 33342 (1 μM, Thermo Fischer Scientific, Waltham, MA, USA) was used for nuclear staining.

Thirty-three fields per well were captured randomly. An average of 825 cells were analyzed per well.

## ROS measurement assay

Quantification of superoxides was performed using 10 μM dihydroethidium (DHE; Thermo Fischer Scientific, Waltham, MA, USA). This dye, when oxidized in the cytosol, intercalates with DNA and emits fluorescence at 605 nm. DHE was applied similarly to MitoTracker and JC-1. As a positive control, NPCs were induced to produce ROS by a 40 min pretreatment with 3.6 μM antimycin A, a mitochondrial complex III inhibitor that stabilizes semi-quinone radicals and favors the escape of electrons to oxygen, thus forming superoxide anions.

Twenty-five fields per well were captured randomly. An average of 6,800 nuclei were analyzed per well.

## Oxygen consumption measurement

Oxygen consumption was measured by high-resolution respirometry using an Oroboros O2k Oxygraph at 37 °C. DataLab software (Oroboros Instruments, Innsbruck, Austria) was used for data acquisition and analysis. NPCs were enzymatically detached from the plate, diluted in culture medium, and seeded to the Oroboros at a concentration of $1 \times 10^6$ cells/mL. The routine oxygen consumption of cells, measured before the addition of modulators of mitochondrial function, was determined after stabilization of the steady state of oxygen consumption for 10–15 min. Subsequently, ATP synthesis was inhibited with 2 μg/mL oligomycin. Oxygen consumption related to oxidative phosphorylation coupled to ATP synthesis was determined by the difference between routine respiration and oligomycin-insensitive respiration.

To uncouple oxidative phosphorylation, the protonophore carbonyl cyanide p-trifluoromethoxyphenylhydrazone was titrated. The resulting maximum oxygen consumption flux value was established as the maximum capacity of the mitochondrial

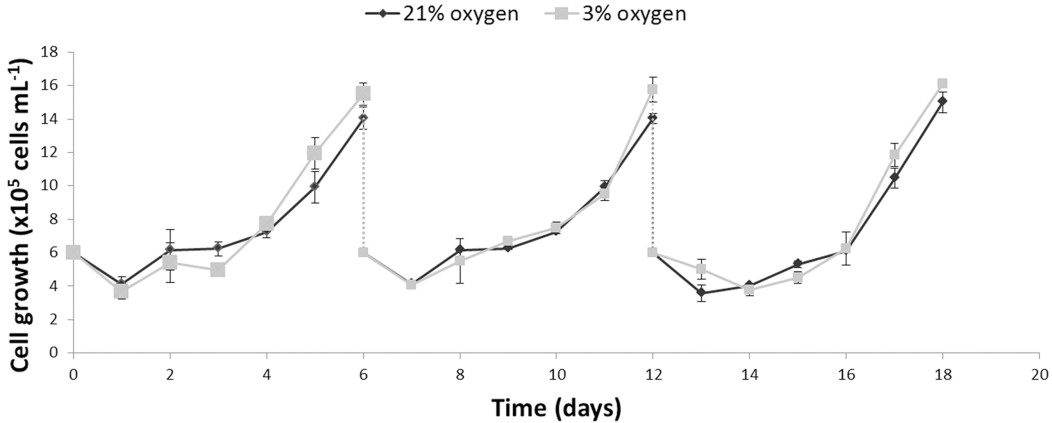

**Figure 1** Comparison of the growth profiles of NPCs cultured either in physioxia (gray line) or normoxia (black line) over an 18-day period, including passaging on days 6 and 12.

electron transport system. Finally, the nonoxidative phosphorylation oxygen flux was determined by blocking the electron transporting system with 1 μg/mL antimycin A. The residual oxygen flux represents oxidases in the cell sample.

### Antioxidant enzymatic activity assay

The activities of the antioxidant enzymes superoxide dismutase (SOD) and glutathione peroxidase (GPx) were measured using commercial kits (cat. 19160 and CGP1, respectively; Sigma-Aldrich, St. Louis, MO, USA). Briefly, the SOD colorimetric assay (440 nm) determines the presence of superoxide radicals in a tetrazolium-coupled reaction. The GPx assay measures NADPH depletion (340 nm). The SOD and GPx enzymatic activities were calculated according to the kit instructions.

To induce ROS production and a possible increase in antioxidant enzymatic activity, the cells were treated with antimycin A, as described previously. Antimycin A showed no effect on apoptosis during the time of treatment (Fig. S1).

### Statistical analysis

The unpaired *t* test was used to compare average differences between two groups. In the case of multiple variable comparisons, one-way analysis of variance was used with the Bonferroni post-test. The null hypothesis of equality between averages was refuted if $p < 0.05$ (*), $p < 0.01$ (**), or $p < 0.001$ (***). Means and standard errors of the mean were plotted.

## RESULTS

### Growth and glycolytic metabolism of human NPCs are not altered in physioxia

The growth kinetics (Fig. 1) as well as glucose uptake and lactate production (Fig. 2) of NPCs in physioxia (3%, v/v) and normoxia (21%, v/v) were similar, indicating that impaired cell growth or a metabolic shift from oxidative respiration to glycolysis (Warburg Effect: *Warburg, 1956*) does not occur in the normoxia-physioxia transition.

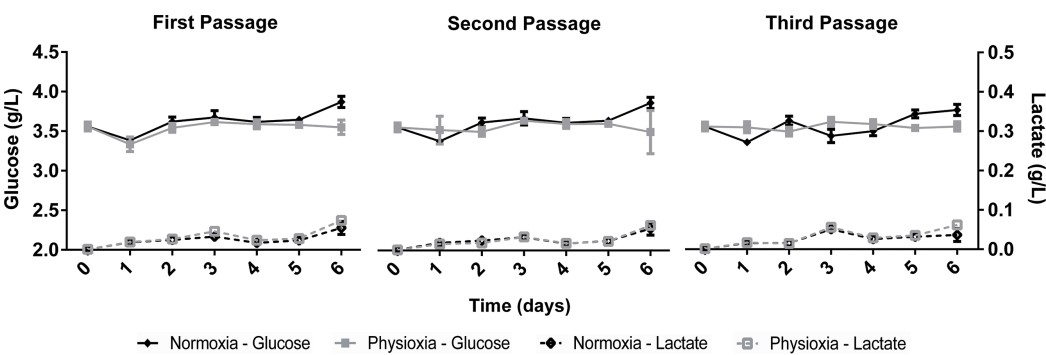

**Figure 2** Glucose uptake and lactate concentration in the supernatant of NPCs grown in physioxia (gray lines) and normoxia (black lines) throughout the first, second, and third passages.

## Mitochondrial content and $\Delta\Psi_M$ are altered in NPCs grown in physioxia

Mitochondrial labeling was performed using the MitoTracker probe, which is internalized by active mitochondria. Fluorescence intensity measurements indicated less mitochondrial content in physioxia-grown NPCs (Figs. 3A and 3B). As a first approach to evaluate mitochondrial function, we quantified $\Delta\Psi_M$ as an indicator of the proton motive force. An increased difference of $\Delta\Psi_M$ can be directly related to a strict commitment to ATP formation by FoF$_1$ATP-synthase and/or to decreased proton leakage to the mitochondrial matrix (*Jastroch et al., 2010*). The data showed increased $\Delta\Psi_M$ in physioxia-grown NPCs (Figs. 3C and 3D).

## Physioxia-grown NPCs feature an increased mitochondrial respiration capacity

Our data suggest that changes in the mitochondrial physiology are induced by $O_2$ availability (Fig. 3). A possible outcome of improved $\Delta\Psi_M$ during mitochondrial function could be a more efficient oxygen consumption directed to ATP production. To verify the full potential of the mitochondria to destine environmental oxygen to oxidative respiration efficiently, we tested the $O_2$ consumption in NPCs grown previously either in normoxia or physioxia in conditions in which plenty of $O_2$ was available, i.e., atmospheric concentrations, by high resolution respirometry. Our data show higher oxygen uptake rates for NPCs previously cultured at 3% $O_2$ (v/v). The oxygen flux coupled to ATP production in NPCs cultured at 3% $O_2$ (v/v) was approximately 70% higher than that of cells cultured at 20% $O_2$ (v/v) (Fig. 4). Additionally, an increased maximum respiratory capacity of physioxia-grown NPCs was observed, compared to those cultured in normoxia. This observation corroborates our previous results, showing that the NPCs did not suffer from restricted mitochondrial function due to lower oxygen availability. Instead, the physioxia conditions increased the potential of oxidative phosphorylation in NPCs cultured at 3% $O_2$ (v/v).

## NPCs grown in physioxia produce less ROS

To verify whether increased oxygen consumption could cause physioxia-grown NPCs to produce more ROS, due to impartial oxygen reduction caused by electron leakage

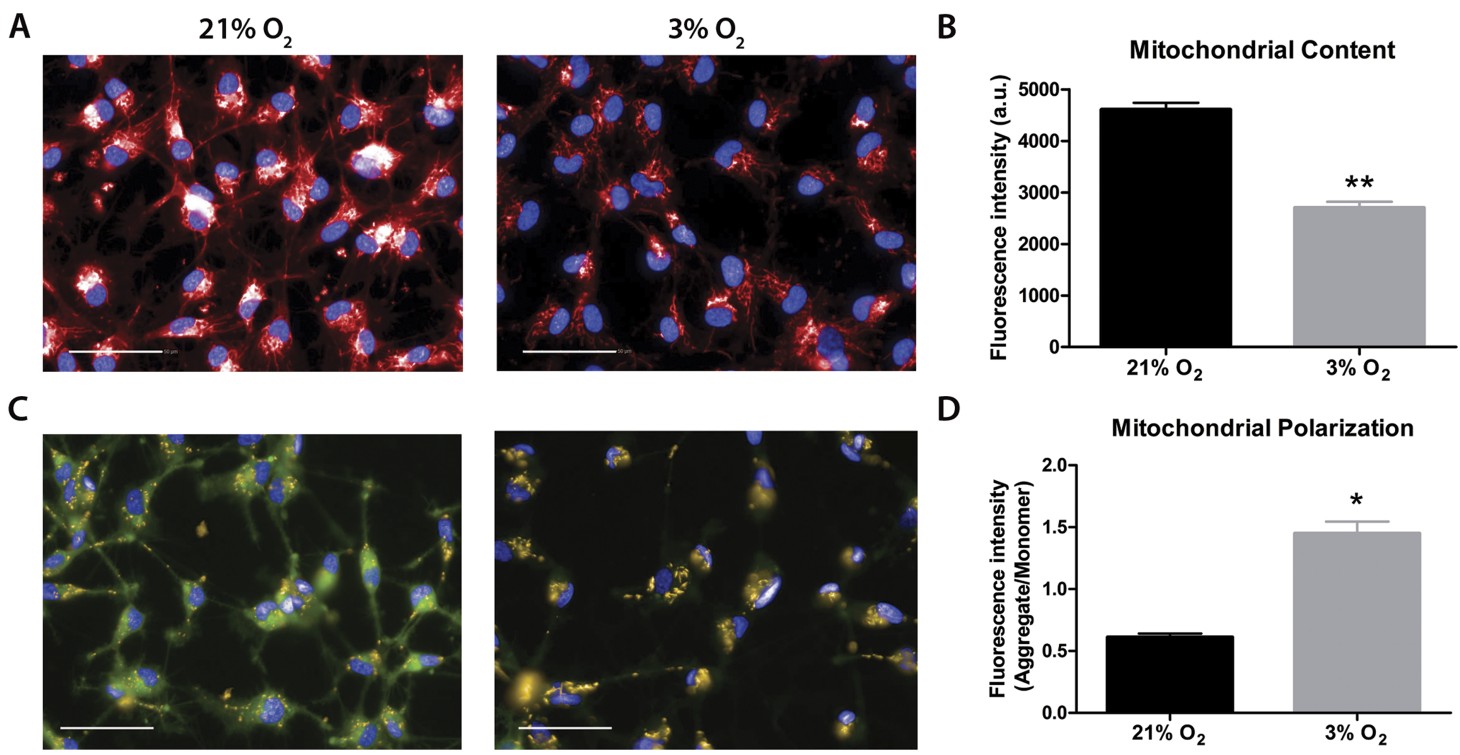

**Figure 3 Evaluation of mitochondrial content by MitoTracker (A and B), in red; and mitochondrial membrane potential by JC-1 (C and D), in green and yellow.** Nuclei are stained in blue by DAPI. $p < 0.05$ (*), $p < 0.01$ (**). Scale bar: 50 μm.

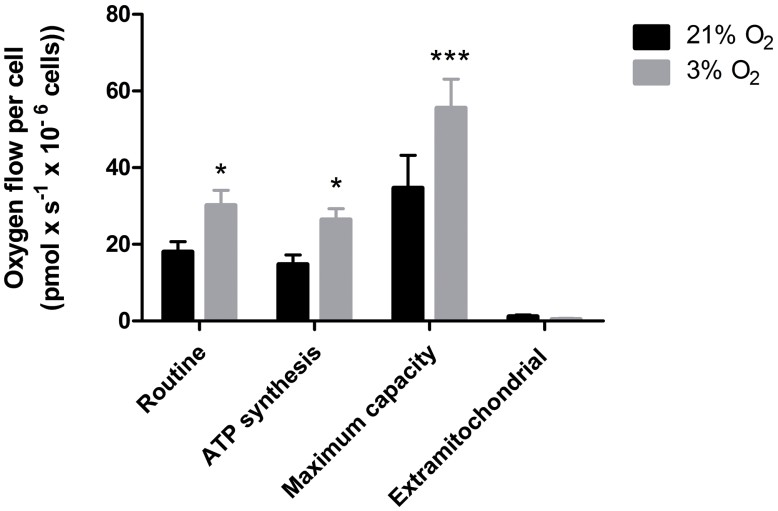

**Figure 4 Oxygen uptake quantification of NPCs grown either in physioxia (gray bars) or normoxia (black bars) by high-resolution respirometry, providing the following parameters: routine respiration, oxygen consumption dedicated to ATP synthesis, maximum mitochondrial capacity, and extramitochondrial respiration.** $p < 0.05$ (*), $p < 0.001$ (***).

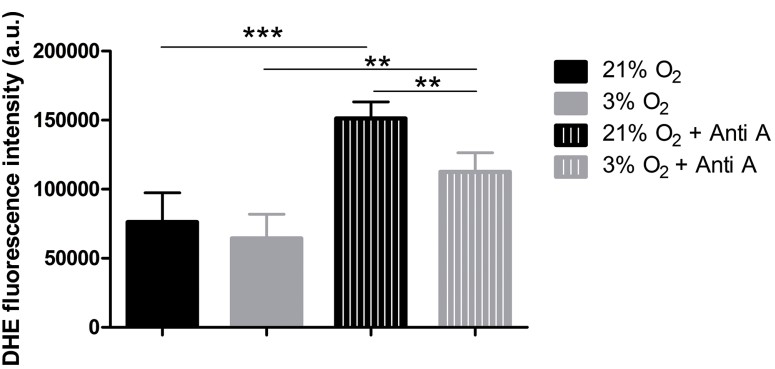

**Figure 5  ROS quantification with the probe DHE.** $p < 0.01$ (**).

from the respiratory chain, we measured ROS production. We used the dye DHE, a permeable probe oxidized in the cytoplasm, which further intercalates DNA and emits fluorescence. As DHE fluorescence was measured in NPCs in their native oxygen environments (i.e., normoxia and physioxia), we observed that, under routine conditions, ROS production was equivalent in both situations (Fig. 5). However, after antimycin A addition, physioxia-grown cells showed lower ROS levels from mitochondrial complex III induced by semi-quinone radical stabilization, compared to normoxia conditions (Fig. 5).

These results suggest that NPCs grown under physiological oxygen are more resistant to redox imbalance. Moreover, this finding also suggests that their antioxidant defense system may be more efficient to scavenge the mitochondrial ROS generated.

## GPx activity is lower in NPCs grown in physioxia

To assess whether the antioxidant defense system of physioxia-grown NPCs is more active under routine conditions, we measured the activities of SOD and GPx in these cells. SOD is responsible for catalyzing the dismutation of superoxide anions in hydrogen peroxide and water. Then, GPx oxidizes intracellular glutathione, reducing peroxide to alcohol and water.

Our data show that NPCs grown either in normoxia or physioxia have equivalent enzymatic activities; however, when treated with antimycin A, the NPCs cultured under lower oxygen concentrations showed a less-pronounced increase of GPx activity in response to the ROS increase, while the SOD activity level was sustained (Fig. 6). This finding indicates that, unlike what was hypothesized previously, NPCs grown in physioxia and treated with antimycin A do not show decreased ROS production due to greater antioxidant activity in the pathway analyzed herein. Instead, GPx, one of the core enzymes of the cellular antioxidant machinery, depicts a decreased response to the ROS increase in these NPCs.

Although the glutathione pathway is not the only one involved in superoxide anion detoxification, it seems to be extremely relevant in NPCs, even surpassing the activity of catalase and peroxiredoxin 3, as shown previously (*Mitozo et al., 2011*; *Xi et al., 2014*). Taken together, these data suggest that, in fact, the antioxidant defense enzymes may be downregulated in physiological oxygen concentrations, compared to atmospheric ones.

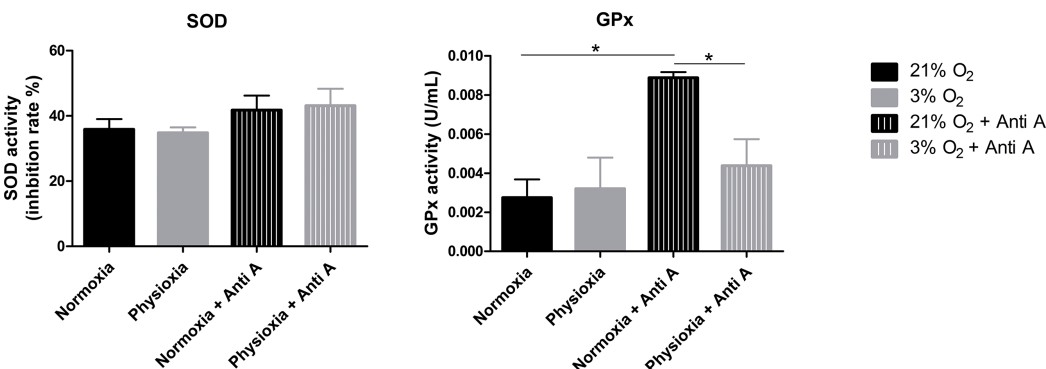

**Figure 6 Enzymatic activity of the antioxidant enzymes superoxide dismutase (SOD) and glutathione peroxidase (GPx).** Activity quantification under routine conditions and after antimycin A treatment, which stimulates maximum ROS production. $p < 0.05$ (*).

## Physioxia-grown NPCs suffer increased DNA damage when exposed to hydrogen peroxide

Although physioxia-grown NPCs produced lower levels of ROS when treated with antimycin A (Fig. 5), our data show that these cells also had a decreased activity of GPx (Fig. 6), one of the main enzymes responsible for peroxide detoxification. Therefore, we investigated whether NPCs cultivated under low oxygen concentrations would be more susceptible to stress. To this end, we quantified DNA damage, an indication of oxidative stress (*Konyalioglu et al., 2013*), in NPCs grown under routine conditions and after hydrogen peroxide treatment.

To quantify the impact of these treatments on DNA damage, the antibody against H2A.X was used to measure altered DNA. H2A.X is phosphorylated when DNA strands break, which signals for DNA repair and cell cycle arrest. Staining of H2A.X appears within the nucleus and can be monitored by both the overall fluorescence intensity and the number of visibly detectable aggregated structures.

Under our routine conditions, we observed that NPCs grown in normoxia or physioxia had equivalent levels of DNA damage (Fig. 7). However, after exposure to hydrogen peroxide, cells cultured under lower oxygen concentrations showed increased DNA damage, which was observed by increased fluorescence intensity both in the whole nucleus and in H2A.X spots (Fig. 7C). Moreover, the population of nuclei containing more than 20 spots decreased; while the whole stained nuclei population, which depicted so many spots that they became indistinguishable from one another, was increased in physioxia-grown NPCs (Fig. 7D).

Increased H2A.X staining of NPCs grown in physioxia and treated with hydrogen peroxide suggests that these cells are more susceptible to exogenous ROS than those cultured in normoxia. Taken together, our results show that higher susceptibility to oxidative stress could be partly due to a lower response of antioxidant enzymes to stress agents in physioxia-grown NPCs, which could contribute to decreased resistance to oxidative stress.

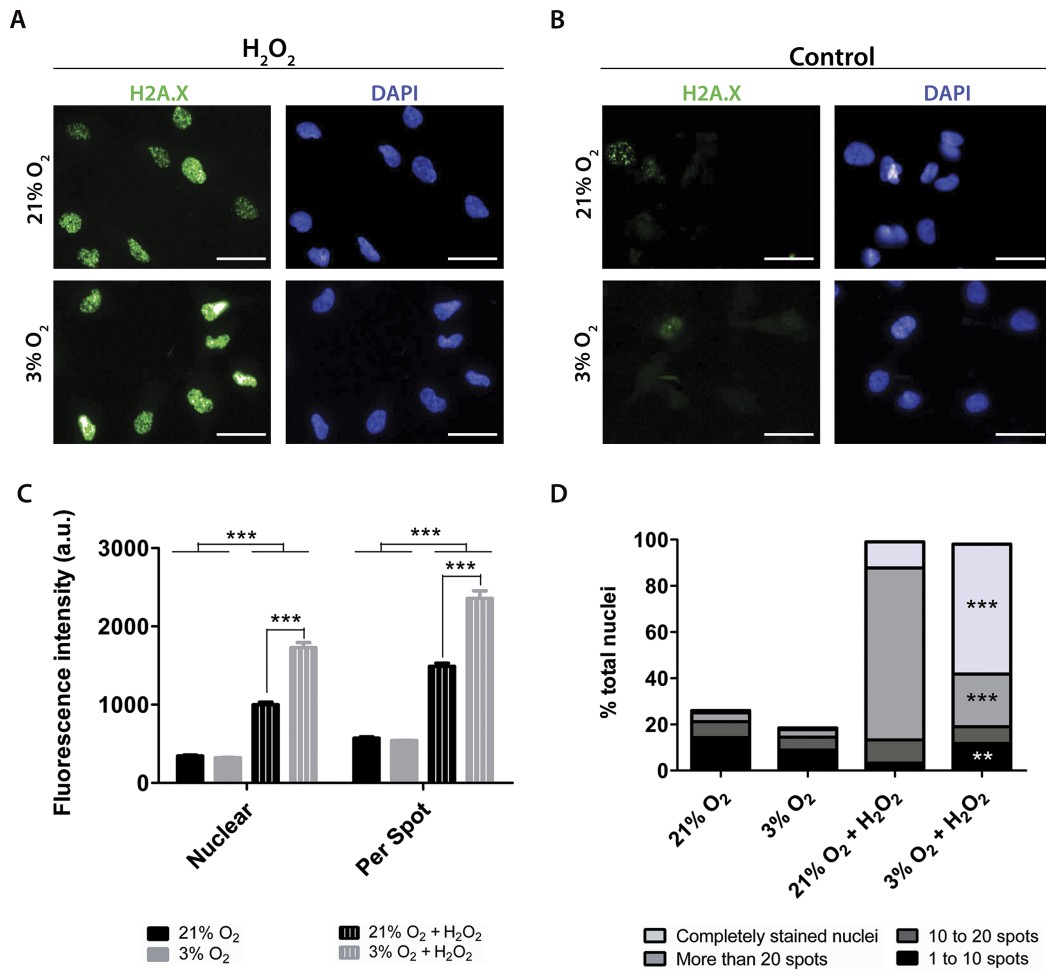

**Figure 7 DNA damage as evidenced by H2A.X (green), either after hydrogen peroxide treatment (A) or under routine conditions (B).** Total nuclei are stained with DAPI (blue). Whole nucleus fluorescence is shown in both oxygen concentrations, after hydrogen peroxide treatment (C) and under routine conditions (D). The fluorescence of the whole nucleus and of the H2A.X spots was analyzed in the four categories. $p < 0.01$ (**), $p < 0.001$ (***). Scale bar: 25 μm.

## DISCUSSION

Evidence of increased susceptibility to exogenous stress agents in physioxia-grown NPCs highlights the importance of culturing human neural cells under low oxygen conditions to better evaluate the effect of drugs, especially the ones known to trigger oxidative stress.

We observed that NPCs in physioxia had reduced mitochondrial content, without changes in morphology (Fig. S2A), which might indicate stress (*Giedt et al., 2012*). Other studies have described lower mitochondrial DNA mass in human carcinoma, which is known to have a decreased oxygen gradient from the border to the center (*Chiba et al., 2013*). Moreover, the NPCs showed no alterations in fusion or fission of the mitochondria, which are linked to the dynamics of these organelles and influence their morphology and amount (*Chan, 2012*).

Although reductions in mitochondrial content can lead to a decrease in ATP produced by oxidative phosphorylation (*Clanton, Hogan & Gladden, 2013*), the lack of bioenergetics

compensation by increased glycolysis and an unaltered growth profile of physioxia-grown NPCs do not indicate reduced mitochondrial activity in these cells. Thus, we evaluated the hypothesis that, instead, there may be improved function of these organelles to maintain ATP production rates necessary for cellular metabolism, even under low $O_2$ concentrations.

Our data reveal that the mitochondria increased the $\Delta\Psi_M$ in physioxia-grown NPCs, indicating a higher activity and mitochondrial efficiency in ATP synthesis. As previous reports have demonstrated that mitochondrial membrane potential increases as oxygen availability is lowered below certain levels (*Tiede et al., 2011*), we believe our system is beneath this threshold. To confirm our hypothesis that increased $\Delta\Psi_M$ could directly affect mitochondrial efficiency and commitment to oxidative respiration in physioxia, we investigated the parameters related to ATP production by measuring the oxygen consumption rates in an atmospheric environment, where oxygen is abundant.

Our high-resolution respirometry results corroborate that, in physioxia, oxygen consumption used for ATP production is increased. Moreover, using this technique, we concluded that a greater $\Delta\Psi_M$ could allow cells to maintain ATP production by oxidative phosphorylation with fewer mitochondria. Thus, we hypothesized that this outcome may reflect an adaptive mechanism conferring a more efficient use of scarce oxygen.

Notably, even with the increased potential respiratory capacity, ROS production was not altered in physioxia compared to normoxia. Notwithstanding, when NPCs were exposed to antimycin A, stimulating an enhanced ROS production by mitochondria (*Chen et al., 2003*), we observed a greater amount of ROS in normoxia-grown NPCs. In this oxygen environment, oxygen availability as a source of ROS production is much higher. Thus, when electron leakage from the respiratory chain is stimulated by antimycin A and the oxygen concentrations increase, superior ROS formation is expected. On the other hand, in physioxia, conditions in which the free oxygen concentration is reduced, the ROS production is decreased even when NPCs are treated with antimycin A, corroborating our hypothesis of greater mitochondrial commitment to ATP production.

Keeping in mind the core importance of the antioxidant enzymes SOD and GPx in the detoxification process of superoxide anions generated by mitochondria in neural cells (even overcoming catalase activity (*Mitozo et al., 2011*)), we evaluated whether the decreased production of these ROS was a consequence not only of increased $\Delta\Psi_M$ but also of improved enzymatic activity.

The SOD activity measurements revealed that this enzyme function was not altered in physioxia, neither under routine conditions nor after antimycin A treatment (Fig. 6). As superoxide anion dismutation in hydrogen peroxide can also occur spontaneously, ROS production can increase after antimycin A treatment even if there are no alterations in SOD activity (*Drose & Brandt, 2008*).

GPx activity was also shown to be very similar between NPCs grown in normoxia or physioxia in routine conditions. However, after antimycin A treatment, cells grown in low oxygen concentrations showed a decreased response, while those cultured in a standard oxygen environment showed a greater increase in GPx activity. While no consensus

currently exists, earlier studies correlating reduced oxygen concentrations to GPx activity indicate decreased activity of this and other enzymes involved in ROS detoxification, such as peroxiredoxin 3 (*Becker et al., 2014*; *Duranton et al., 2012*; *Hidalgo et al., 2014*; *Xi et al., 2014*). Indeed, minor increases of ROS production in response to antimycin A treatment were not enough to generate greater GPx activation in physioxia-grown NPCs.

The importance of GPx peroxide detoxification was investigated once more, measuring the susceptibility of these cells to hydrogen peroxide insults by quantifying the amount of DNA damage, a marker of oxidative stress. Increased DNA damage caused by peroxide, shown previously in other models (*Konyalioglu et al., 2013*; *Mitozo et al., 2011*), was herein reproduced by NPCs in normoxia. These cells showed a greater H2A.X fluorescence intensity and more spots in the nuclei, and the amount of marked cells rose from 25% to 100% after treatment.

Comparing NPCs grown in normoxia vs. physioxia, all parameters indicated a similar amount of DNA damage in routine conditions (without insult). However, after hydrogen peroxide insult, physioxia-grown NPCs showed nuclei with increased H2A.X fluorescence and more spots, indicating increased DNA damage as compared to normoxia-grown NPCs.

These data indicate a greater susceptibility of human NPCs grown in physiological oxygen conditions to insults caused by ROS. This evidence is of great relevance, since models to study disease related to oxidative stress are usually performed under atmospheric oxygen environments, and possibly may not represent actual processes and responses.

Recent studies, using human NPCs differentiated from donors with brain disorders, have shown alterations in mitochondrial membrane potential and oxidative stress, in addition to increased DNA damage, when compared to control NPCs generated from healthy donors; these data indicate phenotype reversibility when classical drugs are used (*Brennand et al., 2014*; *Paulsen et al., 2013*). Just as these studies, many others have been performed under atmospheric oxygen concentrations; thus, the results could be more reliable if the cells had been grown at the same oxygen concentrations found *in vivo*.

Although there are only suggestions of how cell cultivation under physiological oxygen levels could alter disease modeling *in vitro*, cell and metabolic modifications induced by low oxygen environments do occur. Tumor and mesodermal adult stem cell studies have elucidated several mechanisms on how physioxia would alter activation of cellular pathways (*Eliasson & Jonsson, 2010*; *Erler et al., 2004*; *Forristal et al., 2013b*; *Jang & Sharkis, 2007*; *Kubota, Takubo & Suda, 2008*; *Rosova et al., 2008*; *Song et al., 2006*), such as the hypoxia-inducible factor 1 alpha mechanism. On the other hand, studies with pluripotent stem cells (*Forristal et al., 2013a*; *Narva et al., 2013*; *Pereira et al., 2013*; *Zachar et al., 2010*) and derived NPCs have focused on describing how cell differentiation would be affected by oxygen concentrations. In this study, we demonstrated that alterations in physioxia-grown NPCs at the metabolic level not only occur but also lead to a differential response to exogenous oxidative agents (Fig. 8), thus shedding light on the importance of oxygenation control in studies focusing on disease and drug screening assays, in a scalable, efficient, and unbiased manner.

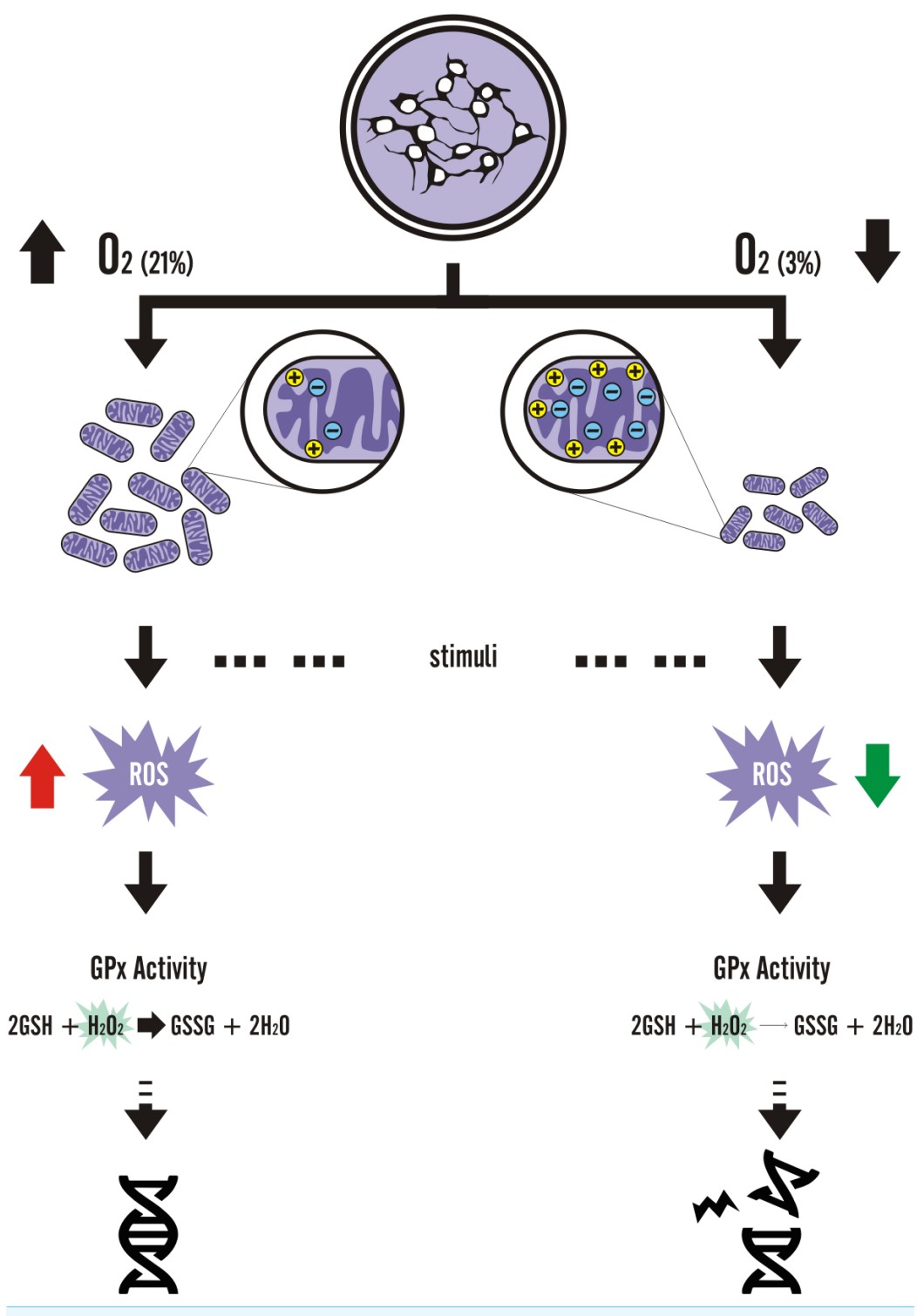

**Figure 8 Graphic summary of the results.**

## ACKNOWLEDGEMENTS

This work is part of the MSc thesis of YL. We thank Michelle Kormann for the support during cell cultivation.

### Funding

This study was funded by the following Brazilian funding agencies: CNPq, FAPERJ, CAPES, FINEP, and BNDES. The funders had no role in study design, data collection and analysis, decision to publish, or preparation of the manuscript.

### Grant Disclosures

The following grant information was disclosed by the authors:
CNPq.
FAPERJ.
CAPES.
FINEP.
BNDES.

### Competing Interests

The authors declare there are no competing interests.

### Author Contributions

- Yury M. Lages performed the experiments, analyzed the data, wrote the paper, prepared figures and/or tables, reviewed drafts of the paper.
- Juliana M. Nascimento conceived and designed the experiments, performed the experiments, analyzed the data, prepared figures and/or tables, reviewed drafts of the paper.
- Gabriela A. Lemos performed the experiments, analyzed the data.
- Antonio Galina, Leda R. Castilho and Stevens K. Rehen conceived and designed the experiments, contributed reagents/materials/analysis tools, wrote the paper, reviewed drafts of the paper.

### Data Availability

The raw data is provided in the Supplemental Information.

### Supplemental Information

Supplemental information for this article can be found online at http://dx.doi.org/10.7717/peerj.1486#supplemental-information.

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
