# Peer review of "Low oxygen alters mitochondrial function and response to oxidative stress in human neural progenitor cells"

_PeerJ, doi:10.7717/peerj.1486_

## Round 0.1 · original submission · Minor Revisions

From the reviews it is evident that the manuscript has received potential scientific interest. The authors are suggested to address minor concerns raised by reviewers 1 and 3. Looking forward to your resubmission.

Reviewer 1 ·

Basic reporting

No comments

Experimental design

No comments

Validity of the findings

No comments

Annotated reviews are not available for download in order to protect the identity of reviewers who chose to remain anonymous.

Reviewer 2 ·

Basic reporting

This study investigates the importance of oxygen levels in neuronal development which will be used for drug screening. The authors showed that altered morphology of human neural progenitor cells and oxidative functions in physioxia condition. They also demonstrate that changes in the mitochondrial physiology in human neural progenitor cells are induced by O2 availability. They showed that human neural progenitor cells cultured in 3% oxygen and exposed to external agents have greater susceptibility to DNA damage.

Experimental design

The authors did an extensive and a thoughtful in vitro analysis to demonstrate the role of oxygen levels in the development of human neural progenitor cells.

Validity of the findings

Overall, this is an interesting study. Interpretations of all the results by the authors are fully supported by the experimental data.

Additional comments

I would like to accept this manuscript for publication.

Reviewer 3 ·

Basic reporting

Looks good

Experimental design

Experimental were well designed and performed.

Validity of the findings

good

Additional comments

This is an interesting and well planned study. Authors have shown that, oxygen levels should be carefully considered for in vitro models of neuronal development and drug screening.
i have few concerns
-Abstract should be reframed, as it is confusing at this moment.
-Authors should include the figures of the NPCs differentiation from human ESCs and their characterization in the manuscript. Which are lacking at this moment.
- Authors should see the effect of antimycin-C on cell death, as authors have shown elevated ROS levels in 21% oxygen (in Fig.5)

---

## Round 0.2 · accepted · Accept

Based on the reviews provided by two experts we found the article to potentially benefit the scientific community. Looking forward to its publication.

Reviewer 1 ·

Basic reporting

This is an excellent study that compares the human neural progenitor cells grown in physioxia and normoxia in terms of growth kinetics, glycolytic metabolism, mitochondrial content, mitochondrial membrane potential, oxygen uptake, ATP production, ROS production, and antioxidant enzymatic activity, thus shedding light on the importance of oxygenation control in studies focusing on disease and drug screening assays.

Experimental design

The authors have done a thoughtful experimental design and the revised article is clearly explained.

Validity of the findings

Authors have evidently demonstrated that alterations in physioxia-grown NPCs at the metabolic level not only occur but also lead to a differential response to exogenous oxidative agents.

Reviewer 3 ·

Basic reporting

Authors attempted to address the concerns raised and satisfactorily addressed the concerns.

Experimental design

No Comments

Validity of the findings

No Comments

Additional comments

Authors did a good job to address the concerns raised.